# Comprehensive Review of Biomarkers for the Treatment of Locally Advanced Colon Cancer

**DOI:** 10.3390/cells11233744

**Published:** 2022-11-23

**Authors:** Jen-Pin Chuang, Hsiang-Lin Tsai, Po-Jung Chen, Tsung-Kun Chang, Wei-Chih Su, Yung-Sung Yeh, Ching-Wen Huang, Jaw-Yuan Wang

**Affiliations:** 1Pingtung Hospital, Ministry of Health and Welfare, Pingtung 90054, Taiwan; 2Department of Surgery, Faculty of Medicine, College of Medicine, National Cheng Kung University, Tainan 70101, Taiwan; 3Department of Surgery, National Cheng Kung University Hospital, Tainan 70101, Taiwan; 4Division of Colorectal Surgery, Department of Surgery, Kaohsiung Medical University Hospital, Kaohsiung Medical University, Kaohsiung 80708, Taiwan; 5Department of Surgery, Faculty of Medicine, College of Medicine, Kaohsiung Medical University, Kaohsiung 80708, Taiwan; 6Department of Surgery, Faculty of Post-Baccalaureate Medicine, College of Medicine, Kaohsiung Medical University, Kaohsiung 80708, Taiwan; 7Graduate Institute of Clinical Medicine, College of Medicine, Kaohsiung Medical University, Kaohsiung 80708, Taiwan; 8Division of Trauma and Surgical Critical Care, Department of Surgery, Kaohsiung Medical University Hospital, Kaohsiung Medical University, Kaohsiung 80708, Taiwan; 9Department of Emergency Medicine, Faculty of Post-Baccalaureate Medicine, College of Medicine, Kaohsiung Medical University, Kaohsiung 80708, Taiwan; 10Graduate Institute of Injury Prevention and Control, College of Public Health, Taipei Medical University, Taipei 11031, Taiwan; 11Graduate Institute of Medicine, College of Medicine, Kaohsiung Medical University, Kaohsiung 80708, Taiwan; 12Center for Cancer Research, Kaohsiung Medical University, Kaohsiung 80708, Taiwan

**Keywords:** locally advanced colon cancer, biomarker, predictive, prognostic

## Abstract

Despite the implementation of global screening programs, colorectal cancer (CRC) remains the second leading cause of cancer-related deaths worldwide. More than 10% of patients with colon cancer are diagnosed as having locally advanced disease with a relatively poor five-year survival rate. Locally advanced colon cancer (LACC) presents surgical challenges to R0 resection. The advantages and disadvantages of preoperative radiotherapy for LACC remain undetermined. Although several reliable novel biomarkers have been proposed for the prediction and prognosis of CRC, few studies have focused solely on the treatment of LACC. This comprehensive review highlights the role of predictive biomarkers for treatment and postoperative oncological outcomes for patients with LACC. Moreover, this review discusses emerging needs and approaches for the discovery of biomarkers that can facilitate the development of new therapeutic targets and surveillance of patients with LACC.

## 1. Introduction

Colorectal cancer (CRC) is the third most common cancer and second leading cause of cancer-related deaths globally. According to the World Health Organization Global Cancer Observatory report, more than 1.9 million new CRC cases occurred in 2020 worldwide, and among them, 1,148,515 were estimated to be diagnosed as having colon cancer [1]. Approximately 10–15% of patients with colon cancer are diagnosed late as having locally advanced colon cancer (LACC), which has poor prognosis [2,3,4,5]. LACC is defined as primary stage T4 colon cancer with direct invasion of surrounding organs or extensive regional lymph node (LN) involvement. According to the American Joint Committee on Cancer (AJCC) staging system [6], LACC is classified into stage IIB/C and stage IIIB/C. Compared with the seventh edition of the *AJCC Staging Manual*, the eighth edition places more emphasis on the poor prognostic features of the degree of tumor invasion despite the presence of fewer positive LNs [7,8]. The degree of LACC invasion can be further divided into T4a stage with visceral peritoneum ingrowth and T4b stage with invasion into nearby tissues or organs [6,8]. According to the Surveillance, Epidemiology, and End Results database [9], the 5-year observed survival rates for colon cancer in stage IIB (T4aN0M0) and stage IIC (T4bN0M0) were 60.6% and 45.7%, respectively. Both stage IIB and stage IIC have poorer prognosis than do stage IIIA (T1-2N1/T1N2a). The 5-year survival rate of patients with node-positive T4 (stage IIIC) disease is less than 40%, which is similar to that of patients with stage IV disease [9,10]. Radical surgery followed by adjuvant chemotherapy remains the mainstay of standard treatment for LACC (Figure 1) [2,5,11,12]. Management of LACC presents surgical challenges because these lesions often extend into adjacent organs or regional LN metastasis involves the root of the main feeding artery [13,14]. In curative-intent surgery, en bloc multivisceral resection (MVR) may be required to ensure adequate resection margins [2,15,16]. Multiple-agent combinations, including 5-fluorouracil (5-FU)/capecitabine, and oxaliplatin (FOLFOX) have been widely used in adjuvant therapy for LACC [17,18,19,20,21]. Adjuvant chemotherapy can reduce recurrence risk and prolong survival in patients with advanced colon cancer, especially those with node-positive or biologically aggressive T4N0 colon cancer [21,22]. In patients with T4b disease or those undergoing R1 or R2 resection, adjuvant chemoradiotherapy resulted in better locoregional control and prolonged disease-free survival (DFS) [23].

Risk factors, including cT4, cN+, and poor or undifferentiated pathology of LACC, significantly reduce the possibility of R0 resection [4,24]. Neoadjuvant chemotherapy can enhance tumor shrinkage, eradicate micrometastases, and prevent tumor cell shedding during surgery. Numerous studies have explored the benefits of neoadjuvant FOLFOX and antiepidermal growth factor receptor (EGFR) therapy for LACC [25,26,27,28]. Neoadjuvant chemotherapy with a FOLFOX regimen could achieve adequate tumor downstaging with acceptable toxicity (Figure 1) [26,28,29,30]. A meta-analysis published in 2020 that included 29,504 patients suggested that neoadjuvant chemotherapy significantly improved overall survival (OS) and DFS without an increase in surgical morbidity compared with upfront surgery followed by chemotherapy for LACC [31]. Hence, the National Comprehensive Cancer Network guidelines recommend neoadjuvant chemotherapy for patients with colon cancer with bulky nodal disease or cT4 status [32].

Approximately 5% of patients with colon cancer have advanced locally unresectable tumors caused by critical organ involvement or direct invasion [33]. Because neoadjuvant chemoradiotherapy is widely administered in locally advanced rectal cancer for its favorable effect on tumor regression [34,35], several studies have investigated its effectiveness in LACC. Our previous study reported that for 34 patients who underwent neoadjuvant chemoradiotherapy and surgery, the pathological complete response and R0 resection rates were 26.4% and 91.2%, respectively [36]. Although these studies had small sample sizes, all of them reported that neoadjuvant chemoradiotherapy is feasible and safe for patients with LACC (Figure 1) [36,37,38,39]. In a single-institutional observational study in which neoadjuvant chemoradiotherapy was administered to 100 patients with unresectable LACC, R0 resection was achieved in 83 patients and grade 3–4 myelosuppression was noted only in 17 patients [39].

Cancer-related biomarkers are beneficial for early detection of cancer and the prediction of prognosis, survival, and treatment response. Colon cancer is a multifactorial malignant disease. Tumor size and its microscopic features, such as the level of aggressiveness; tumor, node, and metastasis (TNM) classification; and lymphatic and venous invasion, have been utilized as fundamental biomarkers for the prediction of prognosis and treatment of colon cancer [32]. In the multistep process model proposed by Fearon and Vogelstein in 1990, the adenoma-carcinoma sequence was identified as the mechanism underlying the development of most colon cancers [40]. Numerous key oncogenes (e.g., *BRAF*, *RAS* [*KRAS* and *NRAS*], and *phosphatidylinositol 3-kinase* [*PIK-3*]), tumor suppressor genes (e.g., *APC*, *TP53*, *DCC*, *TGFβ*, and *SMAD4*), *mismatch repair* (*MMR*) genes, genes regulating microsatellite instability (MSI) and cell cycle, and epigenetic alterations (e.g., aberrant DNA methylation) are involved in colon cancer formation [41,42,43,44]. The differential expression of all these cellular molecules can facilitate the identification of colon cancer biomarkers. Advances in cancer genomic and molecular research over the past 30 years have led to the identification of numerous biomarkers from the tumor, stool, and tissue specimens of patients for colon cancer diagnosis and prognosis prediction [45,46,47]. However, most studies have addressed prognostic instead of predictive biomarkers that may correlate with the survival outcome of adjuvant chemotherapy in stage III colon cancer [18,21,48,49,50,51,52,53]. Therefore, this review article highlights the role of biomarkers in therapeutic approaches and follow-up for patients with LACC and reports emerging trends and findings regarding the biomarkers of LACC treatment.

## 2. Materials and Methods

We systematically searched for studies investigating the role of biomarkers in the management of LACC in PubMed, Cochrane Review, Cochrane Central Register of Controlled Trials, and Clinical Trials.gov databases from their inception until August 2022. No search language or regional restrictions were imposed. Search terms were as follows: “biomarker”, “colon cancer”, “neoadjuvant chemotherapy”, “neoadjuvant chemoradiotherapy”, “locally advanced”, “downstaging”, “adjuvant chemotherapy”, “overall survival (OS)”, and “disease free survival (DFS)”. A thorough manual review of all bibliographies and relevant studies was conducted to identify additional potentially eligible studies.

## 3. Role of Biomarkers in LACC Treatment

### 3.1. Prognostic Biomarkers for LACC

The TNM staging system and adjuvant chemotherapy remain the foundation of prognostication in LACC (Figure 2) [9]. An observational study including 15,489 patients with stage IIB/C disease reported a median survival of 122.6 months for stage IIB/C and the retrieval of ≥12 LNs following adjuvant chemotherapy, 72.5 months for stage IIB/C and the retrieval of <12 LNs following adjuvant chemotherapy, and only 46.5 months for stage IIB/C without chemotherapy [54]. A Dutch study including 10,878 patients with LACC indicated that old age (≥70 year), incomplete resection margin, and nodal positivity status were significantly associated with poor survival [4]. The site of colon cancer development is another key prognostic factor for LACC (Figure 2) [55,56,57,58]. Differences in the embryonic origin divide the colon into left and right sides. The left side refers to the region between the splenic flexure and the upper anal canal, whereas the right side includes the cecum, ascending colon, and transverse colon. In a Taiwanese study including 1095 CRC patients, OS and cancer-specific survival were shorter in right-sided colon cancer compared with left-sided colon cancer for all stages; these survival differences were particularly significant in those with stage III disease [55]. A population-based cohort study including 163,980 patients with colon cancer suggested that left-sided colon cancer was associated with longer OS in stage I, III, and IV disease but shorter OS in stage II disease in patients with left-sided colon cancer than in those with right-side colon cancer [56]. However, another meta-analysis including 66 studies with 1,437,846 patients reported that the longer OS of patients with left-sided colon cancer was independent of stage, race, adjuvant chemotherapy, year of study, number of participants, and quality of included studies [58]. A Taiwanese CRC cohort study indicated that differences in the genomic and metabolomic landscape between right-sided and left-sided CRC may be a potential biomarker [57].

### 3.2. Predictive Biomarkers for LACC Undergoing Neoadjuvant Chemotherapy

For the management of LACC, neoadjuvant chemotherapy can provide better a survival benefit compared with adjuvant chemotherapy, with an acceptable side effect profile [31]. However, a nationwide population-based cohort study only observed this survival benefit in those with locally advanced (T4b) colon cancer but not in patients with T3 or T4a disease. These data suggest that T4b is a critical indicator of the downstaging effect following neoadjuvant chemotherapy [59].

#### 3.2.1. Mismatch Repair Deficiency

Mismatch repair (MMR) deficient (dMMR) cells produce truncated, nonfunctional proteins or exhibit loss of proteins, which can lead to cancer. dMMR occurs in approximately 15% of sporadic CRC cases. With immunohistochemistry, tumors that demonstrate loss of an MMR protein can be classified as dMMR and those with intact MMR proteins can be classified as proficient MMR (pMMR) [60,61]. At the 2020 American Society of Clinical Oncology Annual Meeting, the FOxTROT Collaborative Group, which is the largest phase III trial addressing neoadjuvant FOLFOX chemotherapy with or without panitumumab for 1053 patients with radiologically staged T3-4, N0-2, and M0 colon cancer, demonstrated that neoadjuvant chemotherapy resulted in moderate or greater histological tumor regression in patients with pMMR than in those with dMMR: 23% (128/553) vs. 7% (8/115), *p* < 0.001, and only patients with pMMR could benefit from a decreased risk of relapse at two years [RR = 0.72 (0.52–1.00), *p* = 0.05] but not in dMMR tumours: [RR = 0.94 (0.43 to 2.07), *p* = 0.9] [28]. However, a retrospective study published in 2022 revealed a controversial result: among 52 patients with cT4 colon cancer, the majority of tumor regression grades in both groups were mild [dMMR vs. pMMR: 64.5% (20/31) vs. 47.6% (10/21)] and moderate [dMMR vs. pMMR: 16.1% (5/31) vs. 28.6% (6/21)]. Likewise, more than half of patients with dMMR experienced downstaging comparable to that of pMMR (64.5% vs. 47.6%). Notably, the three-year DFS and OS were 95.2% and 97.0% in patients with dMMR, respectively, compared with 76.2% and 85.7% in patients with pMMR, respectively [62]. MMR status plays a pivotal role in immunotherapy for colon cancer. A study evaluating the efficacy of antiprogrammed cell death 1 (PD1) among patients with advanced dMMR cancer in 12 different tumor types (including colon cancer) elucidated the benefit of immune checkpoint blockade in dMMR-MSI-H colon cancers [63]. In 35 patients with early-stage colon cancers who received ipilimumab with nivolumab, a pathological response was observed in 20 (100%) out of 20 patients (100%; 95% exact confidence interval [CI], 86–100%) in the dMMR group and only 4 (27%) out of 15 patients in the pMMR group [64]. In 2022, Chalabi et al. reported the promising downstaging effect of neoadjuvant immunotherapy in dMMR LACC (Figure 1). Among 112 patients with cT3 or N+ dMMR colon cancer (including 63% clinical T4a or T4b tumors), 95% exhibited a major pathological response, with 67% of them exhibiting a pathological complete response after receiving neoadjuvant ipilimumab plus nivolumab therapy [65]. Immune-checkpoint inhibitors for metastatic colon cancer (mCRC) have demonstrated favorable results for the dMMR/MSI-high (MSI-H) subgroup [66,67].

#### 3.2.2. Excision Repair Cross-Complementing 1, Thymidylate Synthase, and Glutathione S-Transferase pi

In addition to MMR status, the expression of excision repair cross-complementing 1 (ERCC1) has attracted attention for its critical role in the repair of platinum-induced DNA damage. This DNA excision repair protein participates in DNA repair and DNA recombination in human cells. In a study including 70 patients with advanced CRC, the expression of ERCC1 and thymidylate synthase (TS) were investigated as potential negative prognostic factors for FOLFOX neoadjuvant chemotherapy [68]. Another study including 39 patients with advanced CRC indicated that those without the expression of ERCC1 or glutathione S-transferase pi but not TS were more likely to respond to FOLFOX chemotherapy [69].

### 3.3. Predictive Biomarkers for Patients with LACC Undergoing Neoadjuvant Chemoradiotherapy

A growing body of evidence supports the notion that neoadjuvant chemoradiotherapy followed by surgery is a reasonable treatment for LACC [37,38,39]. However, the discovery of predictive biomarkers of neoadjuvant chemoradiotherapy for LACC is still in progress. Both ERCC1 and excision repair cross-complementing 2 (ERCC2) overexpression were associated with poor response to FOLFOX-based concurrent chemoradiotherapy (CCRT) in 14 consecutive patients with cT4b colon cancer, and irinotecan plus 5-FU/leucovorin (FOLFIRI) may be a potential second-line neoadjuvant treatment after FOLFOX-based CCRT failure [70]. In addition to predictive biomarkers, the prognostic indicators of survival for LACC undergoing neoadjuvant chemoradiotherapy have been investigated. Yuan et al. evaluated the efficacy of a neoadjuvant regimen consisting of radiotherapy and fluoropyridine-based chemotherapy for 100 patients with unresectable LACC. This observational study suggested that low differentiation, non-R0 resection, ypT stage (ypT4a-T4b), and advanced ypTNM stage (ypIIb-IIIc) were significantly associated with poor OS and progression-free survival in univariate analysis. However, unlike rectal carcinoma, posttreatment TNM staging is a pivotal prognostic indicator of survival after preoperative CCRT [71]. After multivariate analysis, only differentiation remained an independent prognostic factor for OS. Differences in ypN stage, MMR status, sex, age, and nutritional status were not associated with differences in survival. Notably, patients who achieved a pathological complete response exhibited longer survival than did those who did not achieve a pathological complete response after neoadjuvant chemoradiotherapy. However, the difference was not statistically significant; this was possibly caused by the small sample size [39].

### 3.4. Prognostic/Predictive Biomarkers of Postoperative Adjuvant Chemotherapy for LACC

R0 resection is crucial for curing LACC. Leijssen et al. indicated that even in curative resection, a radial margin of <1 cm and LN involvement were independent predictors of poor DFS [24]. Furthermore, a Dutch study reported that multivisceral resection (MVR) was independently associated with less incomplete resection but not with survival [4]. Likewise, adjuvant chemotherapy in the regimen of oxaliplatin and fluoropyrimidine (FOLFOX or CAPOX) prevents recurrence by eradicating minimal residual disease and has been approved as a standard treatment for stage III colon cancer [21]. The identification of numerous molecular biomarkers for postoperative adjuvant chemotherapy can improve outcomes by allowing the personalization of treatment strategies for patients with LACC.

#### 3.4.1. MMR

The prognostic effect of MMR status on postoperative adjuvant chemotherapy for LACC remains controversial. In a study including 324 patients who underwent radical surgical resection for high-risk stage II or III colon cancer between 2005 and 2008, oxaliplatin-based adjuvant chemotherapy was primarily beneficial for patients with pMMR but may not for patients with tumors that exhibit dMMR [72]. By contrast, a pooled analysis of two randomized clinical trials (NCCTG N0147 and PETACC8), which included 2501 patients with stage III colon cancer, reported that the dMMR phenotype was a favorable prognostic factor for patients with stage III colon cancer receiving FOLFOX adjuvant chemotherapy, and DFS was significantly longer for patients with the dMMR phenotype than for patients with the pMMR phenotype [50]. Notably, two trials are currently ongoing to determine the efficacy of adjuvant anti-PDL1 monoclonal antibodies in patients with stage III dMMR colon cancer (ATOMIC and POLEM) [73,74].

#### 3.4.2. MSI-High

MSI linked to mutations in MMR genes refers to genetic hypermutability (predisposition to mutations) that results in variations in MS sequence length or base composition. This is an alteration often caused by dMMR. The MS status of a tumor may be classified as stable (MSS), low instability (MSI-L), or high instability (MSI-H). In general, dMMR is equivalent to MSI-H. [75]. Tumors with pMMR–MSI-L signature have a lower tumor mutation burden (<8.24 mutations per 10^6^ DNA bases), and those with dMMR–MSI-H signature have a high mutation burden (>12 mutations per 10^6^ DNA bases) [76]. In sporadic colon cancer, MSI is more common in stage II (approximately 20%) and III (12%) tumors than in stage IV tumors (4%) [77]. However, for patients with stage III colon cancer receiving FOLFOX adjuvant chemotherapy, the prognostic role of MSI versus MSS tumors has also been reported. Compared with patients with MSI-L/MSS colon cancer, those with MSI-H colon cancer exhibited no significant differences in five-year DFS and OS [12,18], and the MSI/dMMR phenotype was associated with better survival after relapse than did the MSS/pMMR phenotype in patients with stage III colon cancer after adjuvant chemotherapy [49]. In patients receiving 5-FU treatment, MSI-H was a more reliable favorable prognostic biomarker for relapse-free survival and OS in stage II colon cancer but not in stage III colon cancer [19]. This finding is consistent with those of previous studies [48,78,79].

#### 3.4.3. Epidermal Growth Factor Receptor Expression

Epidermal growth factor receptor (EGFR) is a 170-KDa transmembrane tyrosine kinase. The EGFR/RAS/RAF/MEK/MAPK pathway plays a crucial role in the occurrence, invasion, and metastasis of colorectal cancer [80,81,82]. Huang et al. demonstrated that in 144 patients with stage III colon cancer who underwent radical resection and adjuvant chemotherapy with the FOLFOX regimen, positive EGFR expression and an abnormal postoperative serum carcinoembryonic antigen level were significantly associated with postoperative relapse. Positive EGFR expression was reported to be a significant independent negative prognostic factor for DFS and OS [17]. Moreover, EGFR expression was a prognostic factor for patients with stage III CRC receiving metronomic maintenance therapy [83].

#### 3.4.4. KRAS and BRAF

*KRAS* mutation occurs in 15–35% of localized colon cancer cases, and *BRAF* mutation is a relatively rare event that occurs in 8–10% of patients with localized colon cancer. Both appear to be associated with decreased DFS, survival after recurrence (SAR), and OS [49,84,85,86]. In a pooled analysis including patients with stage III colon cancer receiving adjuvant FOLFOX, *BRAF* or *KRAS* mutations were significantly associated with shorter time to recurrence (TTR), SAR, and OS in those with MSS but not in those with MSI [49]. Among patients with recurrent stage III colon cancer after oxaliplatin-based adjuvant chemotherapy, mutations in *BRAF* were significantly associated with poor SAR [86,87]. Poor SAR for tumors with *BRAF* or *KRAS* mutations was more strongly associated with distal cancers [86].

#### 3.4.5. Phosphatidylinositol-4,5-Bisphosphate 3-Kinase Catalytic Subunit Alpha

Phosphatidylinositol-4,5-bisphosphate 3-kinase catalytic subunit alpha (*PIK3CA*) encodes the p110α catalytic subunit of PI3 kinase. *PIK3CA* mutations are present in approximately 17% of colon cancers and result in the constitutive activation of the kinase and downstream AKT pathway [88]. Patients with *PIK3CA*-mutated CRC appear to have better survival after receiving aspirin in addition to chemotherapy [89]. In the VICTOR trial, which included 896 participants after primary CRC resection, rofecoxib (COX-2–selective nonsteroidal anti-inflammatory drugs) failed to provide effective outcomes in those with stage II–III tumors with *PIK3CA* mutations compared to those with wild-type *PIK3CA*. However, in patients with tumors with *PIK3CA* mutations, the recurrence rate was lower in the regular aspirin use subgroup. This prospective study further supported the benefit of adjuvant aspirin in patients with *PIK3CA* mutations [88]. A double-blind randomized phase III trial (PRODIGE 50-ASPIK) evaluating aspirin (100 mg/d during 3 years or until recurrence) versus placebo in stage III or high-risk stage II colon cancer with a *PIK3CA* mutation after surgical resection is ongoing [90].

#### 3.4.6. CpG Island Methylator Phenotype

The CpG island methylator phenotype (CIMP) phenotype refers to the hypermethylation state of CpG islands localized in gene enhancer regions and occurs in approximately 18% of colon cancer [91]. In stage III colon cancer, patients with CIMP-positive tumors exhibited poorer OS than did those with CIMP-negative tumors [91,92]. CIMP-positive, MMR-intact colon tumors appeared to benefit most from irinotecan-based adjuvant therapy for stage III colon cancer [92]. However, a large cohort study including 1867 patients with stage III colon cancer treated with oxaliplatin-based adjuvant chemotherapy reported that CIMP positivity was associated with shorter OS and SAR but not DFS [91].

#### 3.4.7. Deleted in Colorectal Cancer Protein

The deleted in CRC (DCC) protein is encoded by *DCC* (chromosome 18q21.2) and is a prognostic factor for patients with stage II and III CRC [93]. Gall et al. reported that for patients with stage II and III CRC, the DCC-positive subgroup responded well to adjuvant chemotherapy. By contrast, in the DCC-negative subgroup, no significant difference between chemotherapy and OS or DFS was reported. Therefore, DCC is likely to be a reliable predictor of response to adjuvant chemotherapy in patients with CRC [94].

#### 3.4.8. ERCC1

ERCC1 is a potential predictive biomarker for the efficacy of oxaliplatin-based neoadjuvant chemotherapy or preoperative CCRT in colon cancer treatment [68,69,70]. ERCC1 overexpression is a key predictor of early failure of FOLFOX-4 adjuvant chemotherapy for patients with stage III CRC. Among analyzed patients, ERCC2 and XRCC1 expression exhibited no predictive role [95].

#### 3.4.9. Circulating Tumor Cells or Circulating Tumor DNA

The persistent presence of postchemotherapeutic circulating tumor cells (CTCs) is a potential powerful surrogate marker for determining clinical outcomes in patients with stage III colon cancer receiving adjuvant mFOLFOX chemotherapy [96]. Moreover, CTCs can be used in real-time tumor biopsy for designing individually tailored therapy against CRC [97]. Likewise, postsurgical circulating tumor DNA (ctDNA) analysis is a powerful tool to detect minimal residual disease and is a promising prognostic marker in CRC treatment [98]. Tie et al. demonstrated that in 96 patients with stage III colon cancer, the estimated three-year RFI was 30% when ctDNA was detectable after adjuvant chemotherapy and 77% when ctDNA was undetectable (HR, 6.8; 95% CI, 11.0–157.0; *p* < 0.001) [51].

#### 3.4.10. Blood Sugar Level

Metabolic syndrome, particularly diabetes, is a major etiological risk factor for the development and progression of CRC [99,100,101]. Increased blood sugar levels may drive cancer cell proliferation and increase CRC resistance to chemotherapy [52,92,102]. Yang et al. demonstrated that in 157 patients with stage III CRC (including 107 colon cancer cases) receiving adjuvant FOLFOX6 chemotherapy and having a fasting blood sugar level of ≥126 mg/dL but not a history of diabetes mellitus significantly enhanced oxaliplatin chemoresistance (Table 1). Hyperglycemia can affect clinical outcomes in patients with stage III CRC receiving adjuvant chemotherapy, and the mechanism of oxaliplatin resistance may be related to the increased phosphorylation of SMAD3 and MYC and upregulation of EHMT2 expression [52].

## 4. Advances in Molecular and Epigenetic Biomarkers with Potential Applications in LACC Therapy

### 4.1. Vascular Endothelial Growth Factor (VEGF), Human Epidermal Growth Factor Receptor 2 (HER2), Hepatocyte Growth Factor (HGF), Tyrosine-Protein Kinase Met (c-Met)

VEGF is the most potent angiogenic growth factor and plays the pivotal role of stimulation of angiogenesis in colon cancer [103]. VEGF is expressed in approximately 50% of CRCs and was correlated with colon cancer progression. Increased VEGF expression was significantly associated with advanced lymph node status and distant metastasis [104]. This role of VEGF is strongly supported by studies showing that inhibition of VEGF with the blocking antibody bevacizumab results in decreased angiogenesis and elimination of cancer growth. FDA approves bevacizumab in combination with chemotherapy for mCRC in the U.S. [105]. Erb-B2 receptor tyrosine kinase 2, also known as *HER2*, is a proto-oncogene located on chromosome 17q21 that encodes a transmembrane glycoprotein receptor with tyrosine kinase activity. Amplification of *HER2* oncogene produces a hyperactivation of mitogenic signals and leads to uncontrolled cell proliferation and tumorigenesis [106]. *HER2* overexpression is present in about 2% of all CRCs, approximately 4% of stage III colon cancer patients [107] and 3–6% of stage IV CRCs. It hampers the efficacy of anti-EGFR-targeted therapy and refers to a negative predictive biomarker in mCRC [108]. The PETACC8 study further suggested that HER2 alterations is a new prognostic biomarker in stage III colon cancer from a FOLFOX based adjuvant trial [107]. HGF and its receptor, c-Met (hepatocyte growth factor receptor), are involved in many important biological processes [109]. HGF/c-Met interaction is associated with HGF-activated colon fibroblast-mediated carcinogenesis of colon epithelial cancer cells [110]. c-MET overexpression indicated a poor survival prognosis and predicted shorter PFS during bevacizumab treatment in patients with stage IV CRC [111]. The subgroup analysis indicated that the prognostic effect of the c-Met level was not associated with disease stages, and the c-Met status could be used to evaluate and predict prognosis in CRC patients [112].

### 4.2. MicroRNAs and Long Noncoding RNAs

MicroRNAs (MiRNAs) and long noncoding RNAs (lncRNAs) are the two major families of nonprotein-coding transcripts. MiRNAs are small, single-stranded, 18–25 nucleotide RNAs, and lncRNAs are more than 200 nucleotides in length [113]. Dysregulation of miRNAs and lncRNAs is highly associated with the initiation and progression of colon cancer [114,115,116,117,118]. In addition, interactions between miRNAs and lncRNAs have been identified in tumor-related vascular processes [119]. Numerous studies have investigated the relationship between miRNA expression and CRC incidence and prognosis. However, the majority of these studies have focused on analyzing the predictive role of miRNAs in early colon cancer development and late metastasis. The *lethal-7* (*let-7*) gene was identified as the first human miRNA and functions as a key regulator in development and cancer [120]. Among 13 types of miRNAs in the let7 family, let-7g is highly associated with cancer development, including CRC [121,122,123]. One study observed a significant increase in the let-7g level in the tumor tissues of patients with CRC and its association with poor chemoresponse and disease progression in CRC [121]. However, Chang et al. demonstrated that the expression level of let-7g was significantly lower in the CRC specimens of a Taiwanese CRC cohort and reported the inhibitory effects of let-7g on migration and invasion in a CRC cell line study [123]. Zhu et al. reported that nine key differentially expressed miRNAs (miR-129, miR-217, miR-125a, miR-375, miR-328, miR-125b, miR-144, miR-194, and miR-486) were associated with OS in colon cancer from the Cancer Genome Atlas (TCGA) database. Depending on the cellular condition in which they are expressed, specific miRNAs can act as tumor suppressors or oncogenes. Among these nine key miRNAs, miR-217 and miR-144 were upregulated and the rest were downregulated [116]. Notably, unlike several newly discovered miRNAs, they were reliable diagnostic biomarkers, which can better identify patients with locally advanced rectal cancer (LARC) [124]. Actual clinical patient data including stage and treatment details are not available in the TCGA database. Thus, the relationship between those nine identified miRNAs and therapeutic effectiveness or prognosis remains unclear in patients with LACC. Another study on 60 colon samples of different grades demonstrated that miR-1299 was negatively correlated to the TNM staging of colon cancer through targeting and inhibiting the expression of STAT3 and was closely related to colon cancer prognosis [114]. In addition, several identified miRNAs (miR-140, miR-215, and miR-494) appeared to mediate chemoresistance in colon cell lines [125,126,127].

Several studies have elucidated the crucial role of LncRNAs in regulating tumor initiation and metastasis [128]. Plasmacytoma variant translocation 1 (PVT1) is an lncRNA located on human chromosome 8q24.21 adjacent to the oncogene *C-MYC*. Its expression level is an independent risk factor for OS in patients with CRC and was highly associated with proliferation and invasion capabilities in CRC cell lines [129,130]. In addition, after examining cancerous and adjacent tissues from 210 patients with CRC, PVT1 expression was observed to increase by 51.4% and was significantly correlated with tumor differentiation, degree of invasion, TNM stage, and lymphatic metastasis [131]. A novel lncRNA cancer susceptibility 21 (CASC21), which is also located in 8q24.21 locus, was identified to be differentially expressed in the bioinformatic analysis of TCGA datasets. Its expression was significantly elevated in patients with colon cancer and tended to be higher in more advanced TNM stages [116]. Likewise, upregulation of lncRNA CASC15 was identified in colon cancer tissues and its expression was significantly correlated with clinical TNM stages [115]. Both CASC15 and CASC21 promoted colon cancer proliferation and metastasis through the activation of the Wnt/β-catenin signaling pathway [115,116]. Notably, HOTAIR, an lncRNA involved in the tumorigenesis of malignant tumors, has been reported to increase the radiosensitivity of CRC cells after knockdown [132], and lncRNA-p21 can enhance CRC radiosensitivity by targeting the WNT/β-catenin signaling pathway [133].

### 4.3. Genomic and Metabolomic Biomarkers

CRC is characterized by its strong genetic basis. Different mechanisms are not completely independent of each other, and different CRC cases may or may not involve multiple mechanisms [57,84,134]. Therefore, different approaches to generate predictive polygenic risk scores (PRS) from genome-wide association studies (GWASs) appear to be efficient strategies for early detection and prevention [134,135]. Metabolomics is another approach that has been increasingly used for biomarker discovery in cancers including CRCs. Given that cancer cells have different metabolic phenotypes from normal cells, alterations in the blood metabolite level observed in CRC have provided new noninvasive biomarkers that can differentiate CRC from healthy individuals [136,137]. Su et al. comprehensively analyzed PRS and blood metabolomic profiles in Taiwanese patients with CRC. The study reported significant differences in driver gene mutation rates between Taiwanese and White individuals. However, left-side CRC and right-side CRC could be differentiated on the basis of blood metabolomic results [57].

### 4.4. Methylation Levels of Long Interspersed Nucleotide Elements

Methylation levels of long interspersed nucleotide elements (LINE-1) has emerged as a key molecular player in colon cancer tumorigenesis and development. The degree of LINE-1 methylation serves as an independent factor for determining oncological outcomes in patients with CRC [42,138,139,140]. Antelo et al. reported that a lower degree of LINE-1 methylation (<65%) was associated with significantly poor OS in early-onset patients with CRC [42]. Another study suggested that LINE-1 hypomethylation of the tumor specimen was independently associated with shorter survival in patients with colon cancer [138]. In addition, among patients with stage III colon cancer treated with radical resection and FOLFOX chemotherapy, LINE-1 hypomethylation status was significantly associated with early postoperative recurrence and poor DFS (Table 2) [140].

## 5. Discussion

CRC is the third most common malignancy worldwide and the second leading cause of cancer death [1]. Among CRC patients, those with LACC have poor prognosis and present surgical challenges because of the direct invasion of adjacent structures or extensive regional lymph node involvement [2]. With the advancement of tumorigenesis, numerous molecular and cellular biomarkers have been discovered for the early detection and prediction of the clinical features, prognosis, recurrence, and metastasis of CRC [44,45,47,64,66,67]. However, few studies have specifically focused on LACC. Among them, the majority of studies have evaluated the efficacy of MVR and chemoradiotherapy [3,5,14,24,36], with limited relevant biomarker studies scattered among different trials [28,29]. To the best of our knowledge, this is the first comprehensive review of biomarkers for LACC with a focus on different treatment phases (Table 1). In our literature search, we found relatively few studies on preoperative chemotherapy or CCRT. The likely reason is that many tumor biomarkers cannot be analyzed and compared without a specimen.

LACC is defined as T4 status both with and without positive regional LN involvement, and it includes advanced stage II and stage III cases. To investigate pathogenic processes or responses to therapeutic intervention for LACC, we included studies examining patients with stage III colon cancer. According to the International Duration Evaluation of Adjuvant Chemotherapy Collaboration [141], patients with stage III colon cancer can be divided into low-risk and high-risk subgroups according to their T and N stages. T1-3N1 cases are considered to have a low risk, accounting for approximately 60% of all stage III cases, with a three-year DFS of 80%. T4 and N2 cases are considered to have a high risk, accounting for approximately 40% of all stage III cases, with a 3-year DFS of 60%. In general, LACC is the collective group of all localized T4 tumors with any N status (Figure 2), and the five-year observed survival rates vary widely, from 60.6% to 12.9%, accounting for T4aN0 and T4bN2b, respectively [9]. Therefore, further subgroup analyses are required to advance our understanding of the real-world relationship between oncological outcomes and biomarkers in patients with LACC.

This study has some limitations. First, several studies included colon and rectal cancers without further subcategories [15,83,95,117,131]; thus, the results of the analysis may not be applicable to LACC management. Second, some studies used small sample sizes [70], which may lead to bias from lack of statistical power. Some biomarkers indicated a potential predictive or prognostic value in the treatment of LACC, but the effect was not statistically significant [39] and was not reported in the PubMed database.

The discovery of an increasing number of molecular biomarkers of LACC may facilitate the development of new therapeutic targets. EGFR expression has been shown to be associated with negative prognostic factors for DFS and OS, and with postoperative recurrence in stage III colon cancer [17,83]. In the FOxTROT trial, neoadjuvant panitumumab, a fully human monoclonal antibody that inhibits EGFR, is still being studied for its oncological effects in LACC patients [28]. Interestingly, LACC with *PIK3CA* mutations appeared to predict better survival after receiving adjuvant aspirin therapy [88,90]. In addition, Chalabi et al. recently demonstrated that 95% of patients with dMMR LACC achieved an unprecedented major pathological response after four weeks of neoadjuvant immunotherapy with nivolumab and ipilimumab [65].

In our study, clinicopathological biomarkers showed consistent reliability in prognostic prediction before chemotherapy in patients with LACC [4,9,24,54,55]. On the other hand, the predictive value of molecular or serum biomarkers appeared to be more effective in certain tumor stage or treatment phases [17,49,51,85,88,91,92,94,95]. Among them, MSI and MMR status could be widely used to predict not only the prognosis of LACC receiving neoadjuvant chemotherapy [28,62] and immunotherapy [65], but also the prognosis of patients receiving adjuvant chemotherapy [72,78,86,87]. Through the widespread availability of DNA sequencing technologies and expanded utilization of PRS [134,135], the incorporation of a wider range of molecular and epigenetic biomarkers in the analysis could lead to more robust biomarker score systems and significantly improve the predictive value for LACC patients at different phases of treatment.

## 6. Conclusions

Among all factors associated with oncological outcomes, R0 resection provides the optimal chance of long-term OS, and TNM staging remains the best reliable tool for prognosis prediction in patients with LACC. Sidedness of colon cancer along with genomic and metabolomic differences may be potential prognostic biomarkers. MMR and MSI status play crucial predictive and prognostic roles in patients with LACC receiving neoadjuvant or adjuvant FOLFOX or immunotherapy. Likewise, the expression of ERCC1 is a significant predictive biomarker for patients with colon cancer undergoing neoadjuvant or adjuvant oxaliplatin-based chemotherapy. However, a universal biomarker that can be applied to all LACC is not yet available. Therefore, strategies to investigate the reliability of various biomarkers at different treatment phases can assist in the assessment and management of LACC.

CTC detection and characterization have always been a promising tool for refining prognosis prediction in patients with colon cancer. In the post-GWAS era, thousands of genetic variants that affect the risk of complex human diseases have been discovered. Nonprotein-coding RNAs appear to be efficient and reliable biomarkers because their expression on tumors is significantly associated with TNM classification, MSI status, radiosensitivity, and even chemoresistance in colon cancers. Moreover, these noncoding RNAs interact with RNA and proteins. Interactions between miRNAs and lncRNAs have been identified in tumor-related vascular processes and may serve as prognostic biomarkers and provide more potential targets for designing personalized treatment against LACC.

## Figures and Tables

**Figure 1 cells-11-03744-f001:**
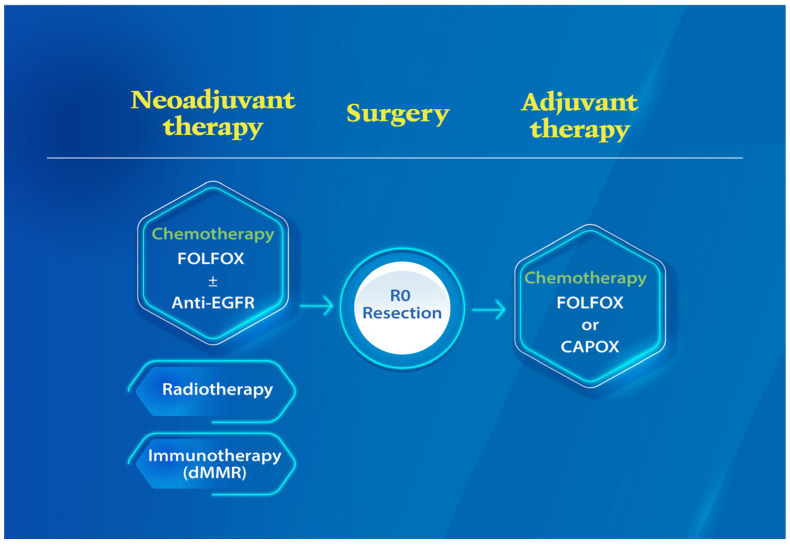
The standard treatment of locally advanced colon cancer. Radical surgery with R0 resection followed by adjuvant chemotherapy adjuvant chemotherapy in the regimen of oxaliplatin and fluoropyrimidine (FOLFOX or CAPOX) remains the mainstay of standard treatment for LACC. Neoadjuvant chemotherapy with FOLFOX regimen can achieve adequate tumor downstaging with acceptable toxicity. Accumulated studies have explored the oncological benefits of neoadjuvant FOLFOX and anti-epidermal growth factor receptor (EGFR) therapy in LACC treatment. On the other hand, neoadjuvant immunotherapy showed promising downstaging effect in dMMR LACC at ESMO 2022.

**Figure 2 cells-11-03744-f002:**
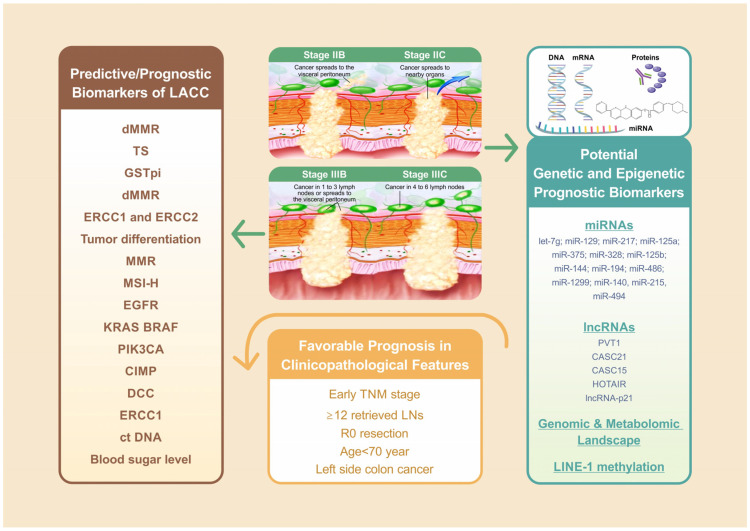
Of all factors associated with LACC treatment outcome, the TNM staging system remains the foundation for LACC prognosis. For stage IIB/C, ≥12 retrieved LNs with adjuvant chemotherapy revealed favorable prognosis compared with those <12 retrieved LNs with or without adjuvant chemotherapy. The further analysis indicated that old age (≥70 year), incomplete resection margin, nodal positivity status are significantly associated with worse survival. Left side colon cancer is another independent favorable prognostic factor for LACC. MMR and MSI status played crucial predictive and prognostic roles in LACC patients receiving neoadjuvant or adjuvant FOLFOX or immunotherapy, and expression of ERCC1 is a significant predictive biomarker for colon cancer patients undergoing neoadjuvant or adjuvant oxaliplatin-based chemotherapy.

**Table 1 cells-11-03744-t001:** Predictive and prognostic biomarkers for LACC in different treatment phases.

LACC Treatment Phases	Biomarkers (Predictive/Prognostic)	Prediction Value (Favorable/Worse)	References
**Clinicopathological features**	TNM staging (Prognostic)	Favorable (early TNM)	[4,9]
	≥12 retrieved LNs (Prognostic)	Favorable	[54]
	R0 resection (Prognostic)	Favorable	[4,24]
	Age ≥ 70 year (Prognostic)	Worse	[4]
	Sidedness of colon cancer (Prognostic)	Favorable (Left side colon cancer)	[55]
**Neoadjuvant** **chemotherapy**	dMMR (Predictive/Prognostic)	Worse	[28,62]
	ERCC1 (Predictive/Prognostic), TS (Prognostic), GSTpi (Predictive)	Worse	[68,69]
**Neoadjuvant** **Immunotherapy**	dMMR (Predictive)	Favorable	[65]
**Neoadjuvant** **Chemoradiotherapy**	ERCC1 and ERCC2 (Predictive)	Worse	[70]
	Tumor differentiation (Prognostic)	Worse (Low differentiation)	[39]
**Postoperative adjuvant chemotherapy**	MMR (Prognostic)	Controversial	[50,72,73,74,86,87]
	MSI-H (Prognostic)	Favorable	[19,48,49,78,79,87]
	EGFR (Prognostic)	Worse	[17,83]
	KRAS and BRAF(Prognostic)	Worse	[49,86,87]
	PIK3CA (Prognostic) ^1^	Favorable	[90]
	CIMP (Prognostic)	Worse	[91,92]
	DCC (Prognostic)	Favorable	[93,94]
	ERCC1 (Prognostic)	Worse	[95]
	ct DNA (Prognostic)	Worse	[51,96,97,98]
	Blood sugar level (Prognostic)	Worse (Fasting blood sugar ≥ 126 mg/dL)	[52]

^1.^ PIK3CA-mutated CC patients appear to have better survival after aspirin therapy in addition to chemotherapy [90].

**Table 2 cells-11-03744-t002:** Molecular and epigenetic biomarkers with potential applications in LACC therapy.

Category	Biomarkers	Predictive/Prognostic Value	References
**Molecular** **Factors**	*VEGF*	Associated with advanced lymph node status and distant metastasis	[103,104,105]
	*HER2*	Negative predictive biomarker in metastatic CRC	[106,107,108]
	*HGF & c-Met*	Poor survival and shorter PFS during bevacizumab treatment in patients with stage IV CRC	[110,111,112]
**miRNAs**	let-7g	Poor chemo-response and disease progression of CRC; migration, invasion in CRC cell lines	[121,123]
	miR-129; miR-217; miR-125a; miR-375; miR-328; miR-125b; miR-144; miR-194; miR-486	Overall survival in colon cancer (Prognostic)	[116]
	miR-1299	TNM staging; colon cancer prognosis (Prognostic)	[114]
	miR-140, miR-215, miR-494	Chemo-resistance in colon cell lines	[125,126,127]
**lncRNAs**	PVT1	Overall survival of CRC patients; proliferation and invasion capabilities in CRC cell lines; tumor differentiation, TNM staging, lymphatic node metastasis (Prognostic)	[129,130,131]
	CASC21	TNM staging (Prognostic)	[116]
	CASC15	TNM staging (Prognostic)	[115]
	HOTAIR	Radiosensitivity of CRC cells	[132]
	lncRNA-p21	CRC radiosensitivity	[133]
**Genomic** **& Metabolomic Landscape**	PRS from GWASs (*KRAS*, *NRAS*, *BRAF*, MSI, Her-2 etc.)	Early detection for CRC	[134,135]
	Blood metabolite levels profile	Early detection for CRC; differentiate left-side CRC and right-side CRC	[57,136,137]
**LINE-1**	Degree of LINE-1 methylation	OS in early-onset CRC patients; survival in colon cancer; postoperative recurrence and DFS (Prognostic)	[42,138,140]

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
