# Peer review of "Comprehensive Review of Biomarkers for the Treatment of Locally Advanced Colon Cancer"

_cells, 2022, doi:10.3390/cells11233744_

Round 1

Reviewer 1 Report

Well written review on biomarkers in locally advanced colon cancer, including staging at initial presentation, tissue- and blood-based assays. 

Line 155: rephrase "our group demonstrated that patients with right-sided colon cancer..." as this has been previously described. 

Line 191: the "only largest" phase III trial is duplicative. 

Line 193: the authors should present the clinical significance of the presented data, including confidence intervals. Include the size of the phase 3 clinical trial, in order to be able to compare with the smaller 2022 retrospective trial described in line 196.

Line 198:  The authors should critically review the clinical data presented. 

Line 275: relationship between MMR and MS status should be discussed here, as dMMR/MSI vs pMMR/MSS phenotypes are described below. 

Line 295 Ref [80]: many of the references are self-referrals. Although not to be faulted, more global critical review of the literature would improve the quality of this manuscript. 

Line 373 Table 1: instead of enumerating the biomarkers, a summary of their value/prediction (e.g., better or worse prognosis) should be included. 

Reviewer 2 Report

Colorectal cancer (CRC) is the third most common cancer and second leading cause of cancer-related deaths globally. Locally advanced colon cancer (LACC) is defined as primary stage T4 colon cancer with direct invasion of surrounding organs or extensive regional lymph node (LN) involvement.

Cancer-related biomarkers are beneficial for early detection of cancer and the prediction of prognosis, survival, and treatment response.

In this manuscript, the authors highlights the role of predictive biomarkers for treatment and postoperative oncological outcomes for patients with LACC. Moreover, this review discusses emerging needs and approaches for the discovery of biomarkers that can facilitate the development of new therapeutic targets and surveillance of patients with LACC.

The figures in this manuscript were quite nice, however the text and logic in this manuscript were not well organized. I have a few concerns:

1)      The title is too wide. This article is just analyzed several biomarker for LACC treatment. Other function of biomarker, such as predict the metastasis potential, the emerging biomarker for drug development, et al, were not mentioned.  The authors should highlight “treatment” or even “adjuvant chemotherapy” in title.

2)       "Biomarker" is not used specifically. It seems the authors only mean molecular biomarker in this manuscript. Moreover, except the molecular biomarker, cell paradigm such as special cell population enrichment, or physiological phenotype such as blood vessel abnormal, could also be defined as biomarker. The authors should indicate “molecular biomarker” in the manuscript.   

3)      Early detection is one of the most important steps for cancer prevent and treatment. However in the current version of this manuscript, early detection have only a few text. Colon cancer is the most well studied cancer type for early detection. There're several commercial available colon cancer early detection kits, such as "Cologuard" in US. Can the authors analyze something about biomarkers in early detection and the commercial status of these kinds of early detection kits? I think biomarkers for early detection should be an independent section in this manuscript. 

4)       There're several colon cancer targeted drugs by direct targeting the special biomarker in colon cancer, such as Bevacizumab for VEGF targeting, Cetuximab for EGFR targeting and Encorafenib for BRAF targeting. Lots of them are 1st line options. However, in this manuscript, this part was totally ignored. The authors didn't mention these biomarker targeting drugs at all. I think the authors should demonstrate these biomarker target drugs in an independent section.     

5)       As mentioned above, VEGF/VEGFR are one of the most important biomarker for colon cancer, either in Prognostic or in drug design. The authors totally ignored VEGF/VEGFR biomarker in this manuscript. Other important biomarkers, such as Her2, HGF, c-Met were not mentioned in this manuscript, either. 

6)      In the abstract, the authors mentioned that the manuscript discussed “approaches for the discovery of biomarkers that can facilitate the development of new therapeutic targets”. Where’s this discussion? If the authors meant the last two paragraphs of the discussion section, I don’t think this is the “approaches for the discover of new biomarkers”, it just subpopulation of CRC patient analysis.  

7)      The font are not consistent. For example, title of 3.4.3 use normal font while the title of 3.4.5 use italic. Please check and correct them.
